# Sensitivity Intensified Ninhydrin-Based Chromogenic System by Ethanol-Ethyl Acetate: Application to Relative Quantitation of GABA

**DOI:** 10.3390/metabo13020283

**Published:** 2023-02-16

**Authors:** Haixing Li, Lingqin Wang, Lijuan Nie, Xiaohua Liu, Jinheng Fu

**Affiliations:** 1State Key Laboratory of Food Science and Technology, Nanchang University, Nanchang 330047, China; 2Sino-German Joint Research Institute, Nanchang University, Nanchang 330047, China

**Keywords:** ninhydrin-based chromogenic system, colorimetry analysis, relative quantitation, gamma-aminobutyric acid, amino acids

## Abstract

Gamma-aminobutyric acid (GABA) is a functional metabolite in various organisms. Herein, a sensitivity intensified ninhydrin-based chromogenic system (SINICS), achieved by ethanol and ethyl acetate, is described for the reliable relative quantitation of GABA. A 2.9 mL SINICS kit comprises 1% ninhydrin, 40% ethanol, 25% ethyl acetate, and 35 μL 0.2 M sodium acetate buffer (pH 5.0). In practice, following the addition of a 0.1 mL sample to the kit, the chromogenic reaction is completed by heating at 70 °C for 30 min. The kit increased the color development sensitivity of L-glutamic acid and GABA, with the detection limits being reduced from 20 mM and 200 mM to 5 mM and 20 mM, respectively. The chromophore was stable for at least 2 h at room temperature, which was sufficient for a routine colorimetric analysis. The absorbance at 570 nm with the deduction of background directly represents the content of amino acid. For a proof-of-concept, the SINICS was adopted to optimize the GABA fermentation process of *Levilactobacillus brevis* CD0817. The results demonstrated that SINICS is an attractive alternative to the available ninhydrin-based colorimetric methods.

## 1. Introduction

As a natural four-carbon metabolite common in various organisms, GABA is mainly produced from the decarboxylation of L-glutamic acid catalyzed by intracellular glutamic acid decarboxylase. In mammals, GABA acts as the chief suppressive neurotransmitter in the central nervous system [1,2,3]. Moreover, GABA displays a variety of other physiological functions, such as sedation, diuretics, hypotensive activity, and memory improvement [4,5,6]. Therefore, GABA has been widely incorporated in the food and pharmaceutical industries as a bioactive factor [7,8,9].

The lactic acid bacterial production of GABA has garnered an increasing attention, because lactic acid bacteria are generally regarded as safe [10,11,12]. However, the production process involves many steps, including screening and identification of GABA-producing strains, optimizing and monitoring of the fermentation process, and the purification of GABA. These steps are frequently interrupted by the arduous task of analyzing numerous samples [13,14,15]. The prevailing methods for assaying GABA include an amino acid analyzer [16], high-performance liquid chromatography (HPLC) [17], and gas chromatography [18]. Although targeting for absolute quantitation, these methods require expensive apparatus or rigorous sample preprocessing steps [19]. In many cases, however, absolute quantitation is not mandatory. In other words, a relative quantitation suffices to meet the associated analysis. Researchers have always been seeking a practical, high-throughput, and cheap method for the relative quantitation of GABA.

The construction of a ninhydrin-based amino acid analysis method is the focus of this study due to its simplicity and high-throughput [20,21]. An α-amino acid reacts with ninhydrin in the acid environment to give a characteristic purple diketohydrindylidenediketohydrindamine, known as Ruhemann’s purple [22]. The proportional relationship between Ruhemann’s purple and amino acid concentration is the basis for the assay [23,24]. A few ninhydrin-based colorimetric methods have been developed and successfully applied to analyze GABA [17,25,26]. However, these colorimetric methods have suffered from poor sensitivity or complex experimental operations [27,28].

Over the past decades, many efforts have been made to seek a convenient and stable reagent system in order to improve the sensitivity of a ninhydrin-based method. Some chromogenic parameters, such as reactant concentration, buffer system, temperature, and heating time, have been proven to exert a significant impact on the formation of Ruhemann’s purple [29,30]. A few organic solvents, for instance, dimethyl sulfoxide or ethanol solution containing cadmium, could improve the sensitivity of the reactions. Recently, an ethanol-ethyl acetate reagent was applied to separate GABA from the fermentation broth, and shows a hyperchromic effect on the GABA-ninhydrin reaction [31,32]. It is then necessary to standardize the reagent via optimizing the key parameters affecting color development so as to facilitate its popularization.

In this work, a SINICS, mainly resulting from the contribution of ethanol and ethyl acetate, was established for the relative quantitation of GABA. First, the formulation of the reagent, pH, and incubation temperature and time, were optimized to maximize the sensitivity and stability of SINICS using L-glutamic acid and GABA as chromogenic objects. Second, the practicability of SINICS was validated by optimizing the GABA fermentation process of *Levilactobacillus brevis* CD0817. Finally, the universality of SINICS was confirmed by applying it to several amino acid representatives classified by their acid-base properties. The data suggest that SINICS is a promising high-throughput method.

## 2. Materials and Methods

### 2.1. Materials

The ultrapure water was prepared by Milli-Q Ultrapure Water System (Millipore Corporation, Billerica, MA, USA). GABA, L-glutamic acid, L-alanine, L-serine, and L-lysine were obtained from Aladdin Biochemical Technology Co., Ltd. (Shanghai, China). Ninhydrin was purchased from the Sinopharm Chemical Reagent Co., Ltd. (Shanghai, China). Yeast extract was purchased from Angel Yeast Co., Ltd. (Wuhan, China). The reagents used for HPLC were HPLC-grade. All other chemicals were of analytical grade. *L. brevis* CD0817 is a previously identified GABA producer [6,33].

The 0.8 M sodium acetate buffer (pH 5.8) was made by dissolving 109.2 g sodium acetate trihydrate and 40.0 mL acetic acid (1 M) to 0.8 L ultrapure water, followed by adding ultrapure water to a final volume of 1 L. The 0.2 M sodium acetate buffer (pH 5.0) contained 17.2 g sodium acetate trihydrate and 73.0 mL acetic acid (1 M). An aliquot of 2.9 mL SINICS comprised 1% ninhydrin, 40% ethanol, 25% ethyl acetate, and a 35 μL 0.2 M sodium acetate buffer (pH 5.0). The 0.1 M solutions of L-glutamate acid, GABA, L-alanine, L-serine, and L-lysine were individually prepared using ultrapure water, and diluted to 10 mM as required. The borate buffer was manufactured by adding 4.9 g boric acid into 100.0 mL of water, adjusting the pH to 10.4 with NaOH, and then diluting it to 200.0 mL. The amino acids derivatization reagent was made by dissolving 10 mg *o*-phthalaldehyde in 2.5 mL acetonitrile, then supplementing 10 μL *β*-mercaptoethanol. The mobile phase of HPLC consisted of solvent A and solvent B at a volume ratio of 4:1. Solvent A was prepared by mixing 9.8 mL of 20.0 mM sodium acetate buffer (pH 7.3) and 0.2 mL triethylamine. Solvent B was acetonitrile.

The seed medium (pH 5.0) for *L. brevis* CD0817 consisted of (g/L): yeast extract, 35.0; MnSO_4_·H_2_O, 0.05; monosodium L-glutamate, 28.0; tween-80, 1.0; and glucose, 10.0. The fermentation medium was (g/L): yeast extract, 35.0; MnSO_4_·H_2_O, 0.05; tween-80, 1.0; glucose, 5.0; monosodium L-glutamate, 33.8; and 650.0 g solid powder L-glutamic acid was added prior to inoculation. Glucose, L-glutamic acid/monosodium L-glutamate, and the other components were individually sterilized at 121 °C for 20 min and mixed together before use.

### 2.2. Methods

#### 2.2.1. Optimization of SINICS

The initial 2.9 mL SINICS only contained 1% ninhydrin and 35 μL 0.8 M sodium acetate buffer (pH 5.8) [31]. To maximize the chromogenic quality of the SINICS, the levels of ethanol, ethyl acetate and sodium acetate buffer, the volume ratio of ethanol to ethyl acetate, temperature, and pH, were individually optimized. To this aim, two amino acids (L-glutamic acid and GABA) were reacted with the reagent as follows: a 0.1 mL sample and 2.9 mL SINICS reagent were added into a screw-top quartz cuvette (Shufu instrument Co., LTD, Wuxi, China) and mixed well, and then incubated in a water bath. The absorbance at 570 nm (*A*_570_) was determined at an interval of 15 min using a UV1200B ultraviolet-visible spectrophotometer (Shanghai Mapada Instruments, Shanghai, China).

#### 2.2.2. GABA Fermentation

In the seed medium, *L. brevis* CD0817 was cultured at 30.0 °C and 100.0 rpm for 5.0–10.0 h until its absorbance at 600 nm (*A*_600_) reached 3.0–6.0, and it was then was used as inoculum. GABA fermentation was initiated by transferring 10.0 mL of the inoculum into a 250-mL Erlenmeyer flask containing 100.0 mL of the fermentation medium, and then was statically kept at 30.0 °C for 48 h [6,7]. Sampling was done every 12 h. The samples were stored at −20 °C until SINICS analysis.

#### 2.2.3. Analysis of Fermentative Samples Using SINICS

A fermentative sample was centrifuged at 12,000 rpm for 5 min. The supernatant was boiled for 5 min and then centrifuged at 12,000 rpm for 5 min, and appropriately diluted with ultrapure water before use. The SINICS test was performed under the optimized chromogenic reaction conditions as described: a 0.1 mL sample and 2.9 mL SINICS reagent were added into a 5 mL centrifuge tube and mixed well, incubated at 70 °C for 30 min, cooled to room temperature, and then the *A*_570_ was determined.

#### 2.2.4. HPLC

An HPLC method was exploited to determine the absolute content of GABA. The fermentation samples were pre-treated as aforementioned. A 20 μL sample was derivatized by reacting with 100 μL of the borate buffer and 20 μL of the amino acids derivatization reagent at room temperature for 5 min. The derivatized sample was analyzed using the Agilent 1200 system (Agilent Technologies Inc., Santa Clara, CA, USA) coupled with an Agilent Eclipse XDB-C18 column (4.6 × 150 mm, 5 μm). The column was eluted by an isocratic mobile phase at a flow rate of 0.8 mL/min, an oven temperature of 30 °C, and a detection wavelength of 338 nm [7,34]. In this work, all of the experiments were repeated at least three times.

### 2.3. The Method of Statistical Analysis

For statistical evaluation, the data were presented as means ± standard deviation, which was calculated using the Analysis ToolPak add-in of Excel 2016. All graphs were plotted using the Origin 2017 software (OriginLab, Northampton, MA, USA).

## 3. Results and Discussion

### 3.1. Optimization of SINICS Chromogenic Parameters

#### 3.1.1. Effects of Ethanol

To determine the role of ethanol, aliquots of 0.1 mL of 10.0 mM L-glutamic acid or GABA solution were individually added into 2.9 mL of the initial SINICS reagents containing different levels (0, 20, 40, 50, and 60%; *v*/*v*) of ethanol. The mixtures were then reacted at 50 °C for up to 120 min. The *A_5_*_70_ values of the reacted mixtures were read with the spectrophotometer at the interval of 15 min. As shown in Figure 1, ethanol significantly fortified the reaction, as the *A_5_*_70_ value increased with ethanol concentration. Few ninhydrin-amino acid chromophores were generated in SINICS with lower levels (0–20%) of ethanol. Given that another enhancer (ethyl acetate) will also be incorporated into the system, 40% ethanol was thus suggested here.

All of the chromogenic reactions were completed within approximately 60 min, since the *A_570_* values peaked at this time (Figure 1). The *A_570_* values were kept almost constant thereafter, which is conducive to ensuring the accuracy and reproducibility of a colorimetric measurement [31,35]. Reportedly, some organic solvents, such as ethanol, methylcellulose solution, and pyridine, could accelerate amino acid-ninhydrin reactions [31,36]. Ethanol has been considered to be a more ideal solvent given its easy availability and good compatibility with water [31].

#### 3.1.2. Effects of Ethyl Acetate

The facilitation of ethyl acetate to an amino acid-ninhydrin reaction cannot be directly evaluated due to its insolubility in water. However, ethyl acetate can be dissolved in an ethanol aqueous solution [31]. In this context, the effects of ethyl acetate were assessed by adding various levels (0, 10, 20, 25, 30, and 40%; *v*/*v*) of ethyl acetate to 40% ethanol-contained SINICS. As presented in Figure 2, the *A*_570_ value was raised with the ethyl acetate concentration, indicating that the chromogenic reaction is promoted by ethyl acetate. The reaction was also completed after 60 min because the *A*_570_ value plateaued at this time. The 25% ethyl acetate should be appropriate because it led to a nearly optimal chromogenic reaction. A composite solvent consisting of 40% ethanol and 25% ethyl acetate was thus determined. The total content of both organic substances in the solvent was 65% (*v*/*v*), which is consistent with not exceeding 70%, as reported [37].

#### 3.1.3. Effects of Ratio of Ethanol to Ethyl Acetate

The above experiments demonstrated that an overall 65% of ethanol plus ethyl acetate facilitates chromogenic reaction. In this section, the total dosage of both organic reagents was fixed at 65%, and the influences of their ratio (35:30, 40:25, 45:20, 50:15, 55:10, 60:5, and 65:0) on the chromogenic reaction were thereafter tested. The results illustrated that the ratio had a marginal effect on *A*_570_ (Figure 3). Nevertheless, 40:25 of ethanol-ethyl acetate maximized the color development of GABA, while it did not maximize that of L-glutamic acid. This phenomenon is helpful to enhance the accuracy of GABA analysis, as the higher the *A*_570_ of GABA is or the lower the *A*_570_ of L-glutamate is, the less interference the L-glutamate has on the assay [31]. Ultimately, 40:25 of ethanol-ethyl acetate was selected in this work.

#### 3.1.4. Effects of Temperature

To identify an appropriate temperature for yielding color, the reaction was performed in the SINICS system under different temperatures (40, 50, 60, 70, and 80 °C) for 120 min. The results showed that both the *A*_570_ value and the reaction completion time were reduced with an increase in temperature (Figure 4). Specifically, 40 or 50 °C gave relatively high *A*_570_ values, but the reaction time was rather long. Under 60 °C, the reaction was completed within 30 min and had a moderate *A*_570_. At 70 or 80 °C, the *A*_570_ value was slightly lower than that at 60 °C, but the reaction could be finished within 15 min, a time comparable to those (10–30 min) of the classical coloring strategies [38]. It seemed that 70 °C is more appropriate than 80 °C because of a relatively high *A*_570_ value. The previous studies used a higher temperature (>85 °C), and even boiling water, for the color development [35,38,39]. However, an excessively high temperature may be not suitable for SINICS, as the resultant chromophore is unstable (Figure 4). After comprehensive consideration of the reaction completion time and *A*_570_, 70 °C was adopted in this study.

#### 3.1.5. Effects of pH

The ninhydrin-amino acid reaction could be performed in a wide pH range [38]. To check the influences of pH on the chromogenic reaction, 35 μL of 0.8 M acetate buffers with different pHs (4.0, 4.5, 5.0, 5.4, 5.8, 6.2, and 7.0) were individually included in the SINICS system. The reaction was conducted under 70 °C for 120 min, and the *A*_570_ was detected every 15 min. Figure 5 shows that the color development was accelerated at a lower pH, due to the fact that the formation of Ruhemann’s purple is acid dependent [38,40]. Nevertheless, the lower the pH, the poorer the stability of chromophore. A mild acid environment (pH 5.0) should be preferable for taking *A*_570_ and chromophore stability into account. The optimal pHs reported by different labs are somewhat different, but generally fall into a mild acid range of 5.0 and 6.0 [38].

#### 3.1.6. Effects of Sodium Acetate Concentration

To investigate the influences of acetate level on chromogenic reaction, 35 μL various concentrations (0, 0.2, 0.4, 0.6, 0.8, and 1.0 M) of sodium acetate buffers (pH 5.0) were individually added into the chromogenic system. The results are summarized in Figure 6. Acetate could either strengthen the reaction or stabilize the chromophore. However, the higher level of sodium acetate, the lower the *A*_570_ value that was achieved. The 0.2 M acetate maximized the color yield, accompanied by a satisfactory chromophore stability [31,35,41].

### 3.2. Stability of Chromophore

The chromophore stability is crucial for colorimetric analysis [4,38]. Upon the completion of color development, the solution was left at room temperature for 120 min, and its *A*_570_ was measured every 15 min. As shown in Figure 7, the chromophore resulting from GABA or L-glutamic acid was almost invariable for at least for 120 min, a duration long enough for colorimetric analysis [13,31].

### 3.3. Chromogenic Sensitivity of SINICS

As the reports indicate [22,36], some organic reagents can accelerate the amino acid-ninhydrin reaction. Here, a reagent of 40% ethanol, 25% ethyl acetate, 1% ninhydrin, and 35 μL 0.2 M sodium acetate buffer (pH 5.0) was developed as a medium to facilitate the reaction. To check the performance of this reagent, the reaction was performed under 70 °C for 30 min using GABA or L-glutamic acid of various concentrations as the reactive substrate. As shown in Figure 8, the detection limits of the SINICS system for GABA and L-glutamic acid were low to 20 and 5.0 mM, respectively. However, the detection limits of the water-based system for both were 200.0 and 20.0 mM, respectively. The SINICS system showed high chromogenic sensitivity relative to the classical one [38,42].

Relative to ninhydrin, *o*-phthalaldehyde and 1,8-diazafluoren-9-one are the two sensitive reagents [43,44,45]. Jones et al. [43] has established a spectrofluorometric method, based on *o*-phthalaldehyde, for rapidly and sensitively quantitating total free amino acids in soil. However, *o*-phthalaldehyde has been used more frequently to derivatize amino acids in the methods with separation step(s), such as HPLC or an amino acid analyzer [16,17,46,47], while 1,8-diazafluoren-9-one is a frequently-used visualizer of latent fingerprints in forensics [44,45]. When analyzing some biological samples enriched in amino acid(s), such as fermentative ones, ninhydrin-based colorimetry is still rather popular.

### 3.4. Validation of SINICS

To date, any ninhydrin-based spectrophotometric method cannot discriminate a specific amino acid from a pool of amino acids due to the fact that ninhydrin can easily react with any amino acids. In other words, the amino acid amount determined by ninhydrin is total [38,43]. Herein, we suggest that the net value of total *A*_570_ minus the background is employed as the indication of GABA content of a sample. Obviously, the key to using SINICS is to ensure that all samples have an approximate level of background, so that the net *A*_570_ value can correctly reflect the GABA content. For a proof-of-concept, the SINICS method was exploited to optimize yeast extract dosage and temperature in the GABA fermentation driven by *L. brevis* CD0817. The fermentation medium contained 0.2 M monosodium L-glutamate; and was added to solid L-glutamic acid at 650 g/L just before the fermentation. The added substrate mainly existed in the form of powder due to its low solubility (about 10 g/L) [17,48]. The detailed experimental operations for the fermentation are presented in Section 2.2.2.

With the initiation of fermentation, dissolved L-glutamic acid was gradually converted into GABA, but was simultaneously compensated by the synchronous dissolution of the powder substrate. As a result, the concentration of GABA was gradually increased with the fermentation, while that of dissolved L-glutamic acid was almost invariable before the powder was depleted. At the later stage of fermentation, owing to the depletion of L-glutamic acid powder, the level of dissolved L-glutamic acid may decrease [6,7,48]. As stated above, in SINICS analysis, GABA content is directly expressed by subtracting *A_570_* from the background control. It is pivotal to ensure that all samples have a similar background so as to make the net *A*_570_ values comparable. To achieve this, we saturated the samples without visible solid substrate with L-glutamic acid powder prior to the SINICS analysis. Another key point is that all samples should be diluted at the same multiple.

In this work, all of the samples were diluted by 400-fold with ultrapure water before the SINICS assay. The fresh fermentation medium (containing 0.2 M monosodium L-glutamate) saturated with L-glutamic acid should be used as the background control. However, the reading of 400-fold dilution of the medium without L-glutamate was negligible (<0.05). For convenience, the aqueous solution of 0.2 M monosodium L-glutamate saturated by L-glutamic acid powder was selected as the control in this work. All the samples also underwent HPLC analysis, as depicted in the Materials and Methods section. As shown in Figure 9, the plot of GABA production determined by SINICS followed that determined by HPLC, suggesting that SINICS could explicate the GABA fermentation process. In fact, the SINICS kit has been successfully used to optimize the whole-cell catalysis process of *L. brevis* CD0817 for the production of GABA [6].

Except for a simple sample dilution, the SINICS method does not involve any extra operations, such as derivatization, filtration, or separation, which are standard to the prevailing methods including HPLC and the amino acid analyzer [27,48,49]. Overall, the SINICS is a simple, convenient, and economical method but can give a satisfactory result, making it an attractive alternative to the available techniques in the related areas.

As with the other spectrophotometric methods, a limitation of SINICS is that this procedure cannot absolutely quantitate the target amino acid because it lacks the ability to distinguish amino acids. We recommend the combination of the SINICS and an absolute quantitative method; that is, the former is exploited to complete optimization/monitoring, and the latter to measure the optimized content of amino acid [6]. Zeng et al. [46] has developed an amino acid analyzer protocol for simultaneously quantifying 17 amino acids in tobacco leaves, in which amino acids were derivatized with ninhydrin. Sancheti et al. [50] has proposed a ninhydrin-based high performance thin layer chromatographic method to simultaneously assay L-glutamic acid and GABA in mice brain. These accurate methods are suitable for determining the final content of amino acid after SINICS optimization.

It would be interesting to identify a reagent that can fortify the color development of a target amino acid, because this reagent may improve the reliability of analysis by reducing the interference of the other amino acids and the sample matrix. Figure 3 shows that 40:25 of ethanol-ethyl acetate maximized the color development of GABA, but did not maximize that of L-glutamic acid, suggesting the possibility of obtaining such a reagent.

### 3.5. Versatility of SINICS

Here, several amino acids (L-alanine, L-glutamic acid, L-serine, and L-lysine) classified according to their acid-base property were selected to evaluate the universality of SINICS. Aliquots of 0.1 mL of 10.0 mM amino acids were individually added to 2.9 mL SINICS kits and then heated under 70 °C for 120 min. As shown, all the chromogenic reactions were completed within 15 min (Figure 10A). Furthermore, all of the reacted solutions showed maximum absorption at 570 nm (Figure 10B), in accordance with the traditional ninhydrin-based methods [38,42]. These data indicated that the SINICS method has potential in assaying other amino acids.

## 4. Conclusions

A SINICS method has been established for the relative quantitation of GABA in this work. A 2.9 mL SINICS kit comprises 1% ninhydrin, 35 μL 0.2 M sodium acetate buffer (pH 5.0), 40% ethanol, and 25% ethyl acetate. The chromogenic reaction is conducted by adding a 0.1 mL sample to a 2.9 mL kit and then incubating it at 70 °C for 30 min. The proposed method includes the features of high sensitivity, high stability, and high-throughput, along with simplicity. The SINICS shows a potential in screening amino acid-producing microbes and optimizing/monitoring their fermentation processes.

## Figures and Tables

**Figure 1 metabolites-13-00283-f001:**
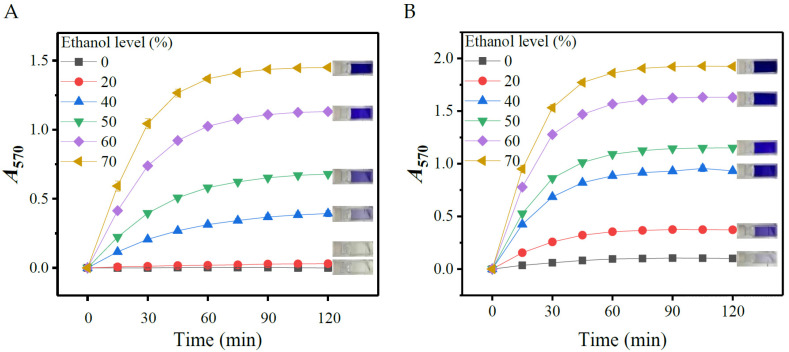
Effects of ethanol level on color yield of gamma-aminobutyric acid (**A**) and L-glutamic acid (**B**). The rightmost part of each diagram shows the reacted mixtures containing different levels (from top to bottom: 70, 60, 50, 40, 20, and 0%) of ethanol. *A*_570_: absorbance at 570 nm.

**Figure 2 metabolites-13-00283-f002:**
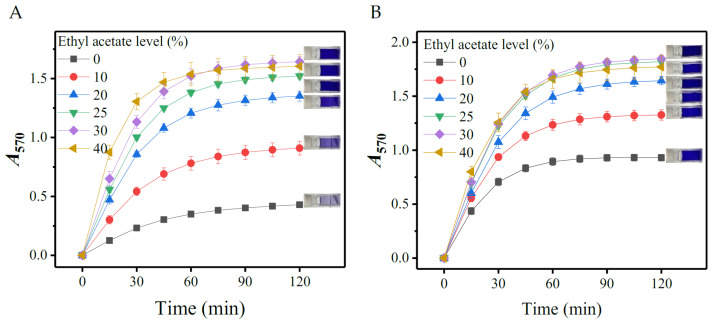
Effects of ethyl acetate level on color yield of gamma-aminobutyric acid (**A**) and L-glutamic acid (**B**). The chromogenic reactions were performed in SINICSs (containing 40% ethanol) supplemented with various levels of ethyl acetate. The rightmost part of each diagram shows the reacted mixtures containing different levels (from top to bottom: 40, 30, 25, 20, 10, and 0%) of ethyl acetate. SINICS: sensitivity intensified ninhydrin-based chromogenic system; and *A*_570_: absorbance at 570 nm.

**Figure 3 metabolites-13-00283-f003:**
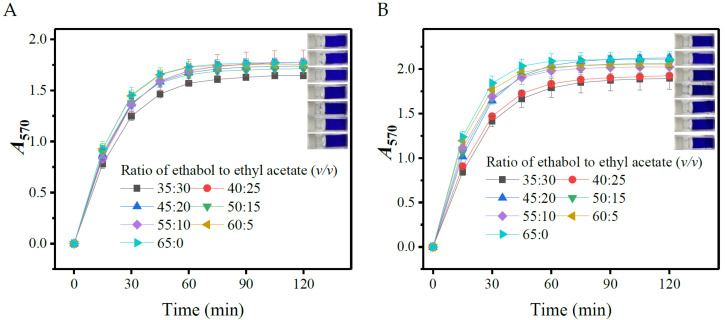
Effects of the ratio of ethanol to ethyl acetate on the color yield of gamma-aminobutyric acid (**A**) and L-glutamic acid (**B**). The rightmost part of each diagram shows the reacted mixtures with different ratios of ethanol to ethyl acetate (from top to bottom: 65:0, 60:5, 55:10, 50:15, 45:20, 40:25, and 35:30). *A*_570_: absorbance at 570 nm.

**Figure 4 metabolites-13-00283-f004:**
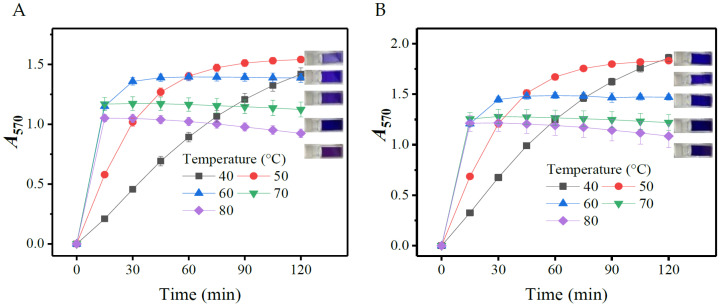
Effects of temperature on color yield of gamma-aminobutyric acid (**A**) and L-glutamic acid (**B**). The rightmost part of each diagram shows the reacted mixtures under different temperatures (from top to bottom: 40, 50, 60, 70, and 80 °C). *A*_570_: absorbance at 570 nm.

**Figure 5 metabolites-13-00283-f005:**
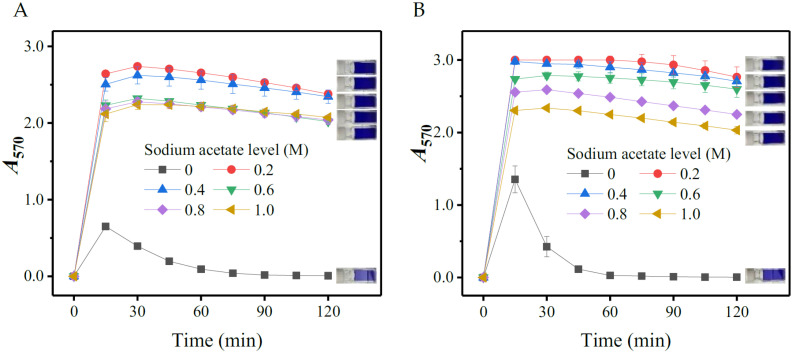
Effects of pH on the color yield of gamma-aminobutyric acid (**A**) and L-glutamic acid (**B**). The rightmost part of each diagram shows the reacted mixtures containing 35 μL of 0.8 M acetate buffer with different pHs (from top to bottom: 4.0, 4.5, 5.0, 5.4, 5.8, 6.2, and 7.0). *A*_570_: absorbance at 570 nm.

**Figure 6 metabolites-13-00283-f006:**
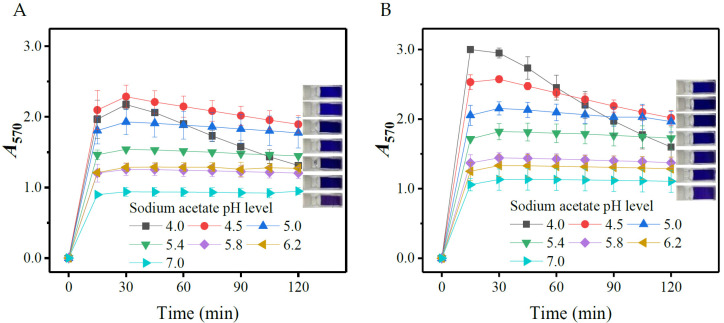
Effects of sodium acetate concentration on the color yield of gamma-aminobutyric acid (**A**) and L-glutamic acid (**B**). The rightmost part of each diagram shows the reacted mixtures supplemented with 35 μL of variable concentrations (from top to bottom: 1.0, 0.8, 0.6, 0.4, 0.2, and 0 M) of sodium acetate (pH 5.0). *A*_570_: absorbance at 570 nm.

**Figure 7 metabolites-13-00283-f007:**
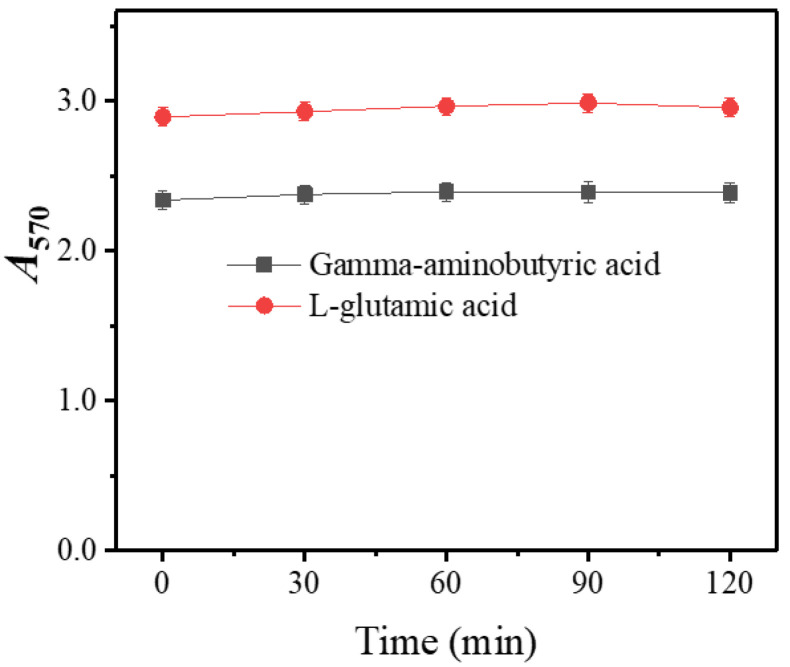
Stability of ninhydrin-based chromophore of gamma-aminobutyric acid or L-glutamic acid. *A*_570_: absorbance at 570 nm.

**Figure 8 metabolites-13-00283-f008:**
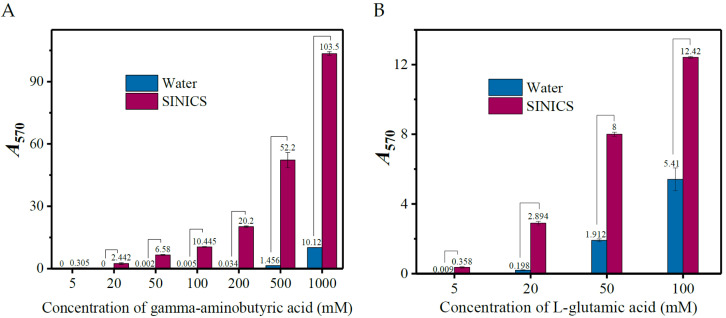
Chromogenic sensitivity of SIANCS to gamma-aminobutyric acid (**A**) and L-glutamic acid (**B**). The control reaction medium used water as a substitute for 40% ethanol-25% ethyl acetate. SINICS: sensitivity intensified ninhydrin-based chromogenic system; and *A*_570_: absorbance at 570 nm.

**Figure 9 metabolites-13-00283-f009:**
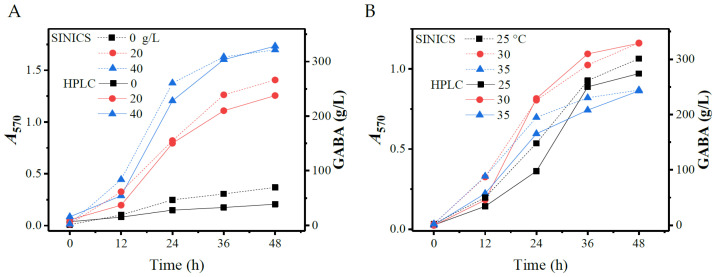
Optimization of GABA fermentation process performed by *Levilactobacillus brevis* CD0817 using HPLC and SINICS. (**A**) effects of yeast extract level on GABA formation; and (**B**) effects of temperature on GABA formation. HPLC: high performance liquid chromatography; SINICS: sensitivity intensified ninhydrin-based chromogenic system; GABA: gamma-aminobutyric acid; and *A*_570_: absorbance at 570 nm.

**Figure 10 metabolites-13-00283-f010:**
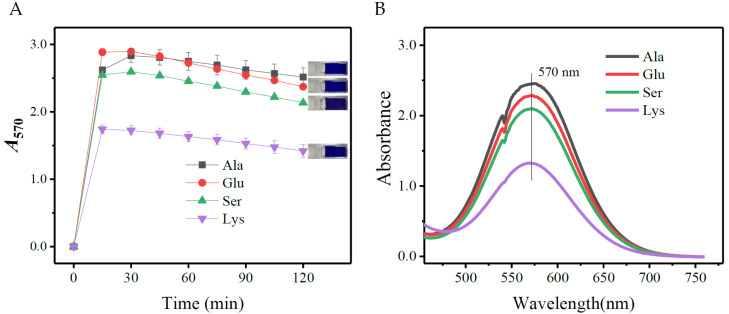
Chromogenic reactions of several amino acids performed by SINICS. (**A**) chromogenic processes; and (**B**) wavelength scans of chromogenic products. The rightmost part of diagram A shows the reacted mixtures of the selected amino acids (from top to bottom: L-alanine, L-glutamic acid, L-serine, and L-lysine). SINICS: sensitivity intensified ninhydrin-based chromogenic system; and *A*_570_: absorbance at 570 nm.

## Data Availability

The data presented in this study are available in this article.

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
