# Peer review of "Sensitivity Intensified Ninhydrin-Based Chromogenic System by Ethanol-Ethyl Acetate: Application to Relative Quantitation of GABA"

_metabolites, 2023, doi:10.3390/metabo13020283_

Round 1
Reviewer 1 Report
In the manuscript titled "Sensitivity intensified ninhydrin-based chromogenic system by ethanol-ethyl acetate: application to relative quantitation of GABA", the authors had developed a sensitive chromogenic system for relative quantitative analysis of GABA. The system is traditionally based on a chromogenic reaction with ninhydrin. But the authors have optimized the formulation of reagent (including the concentration of aqueous ethanol and ethyl acetate), buffer system, pH, and incubation both temperature and time to maximize the sensitivity and stability the chromogenic reaction. An enhanced system was deservedly called “a sensitivity intensified ninhydrin-based chromogenic system (SINICS)”. In my opinion, the presented results of the study are relevant, interesting and within the scope of the Journal. I have just some recommendations to improve the text:
1) As for the readers, the authors’ choice of 40% aqueous ethanol is not obvious or indisputable from the text (see Figure 1), since a 40% ethanol solution diluted with ethyl acetate may be caused a precipitation of some components (or reagents) of the analytical mixture. Moreover, the choice of 60% aqueous ethanol seemed to be more justified, since the absorption value of 60% aqueous ethanol solution was almost three times higher than that of 40% (Figure 1).
2) Although the authors claimed that "All the chromogenic reactions were completed within around 60 min" (part 3.1.1), Figure 1 shows this is not so clear (the absorption value does not yet reach its maximum for this duration). Therefore, calculating the GABA concentration, a certain multiplying factor could be used. The same is true for choosing 25% ethyl acetate instead of 30-35% (see Figure 2).
3) What is the role of ethyl acetate in the chromogenic reaction? Why does ethyl acetate (according to the authors’ statement) promote "amino acid-ninhydrin reaction"? I can find no explanation in the text. Figure 3 rather refutes the requirement of the ethyl acetate addition, but indicates the need to use a higher concentration of ethanol.
4) Regarding to the temperature effect, the authors suggest to use 70°C, although 60°C is obviously better (see Figure 4).
5) Why do the authors believe that pH 5.0 is better than pH 4.5? Over 120 minutes, the absorption values are always higher for pH 4.5 than that for pH 5.0 (see Figure 5).
6) There is a mistype in Abstract: "The kit increases color development sensitivity of L-glutamic acid and GABA, with the reduced detection limit from 20 mM to 5 mM and 20 mM to 5 mM, respectively." According to the information for GABA in part 3.3, the reduced detection limit was from 200.0 mM to 20.0 mM.
7) There are two relevant articles for the subject studied:
a) Y. Zeng, W. Cai, X. Shao, Quantitative analysis of 17 amino acids in tobacco leaves using an amino acid analyzer and chemometric resolution. J. Separ. Sci., 2015, 38 (12), 2053–2058. DOI: 10.1002/jssc.201500090
b) J.S. Sancheti, M.F. Shaikh, P.F. Khatwani, S.R, Kulkarni, S. Sathaye, Development and validation of a HPTLC method for simultaneous estimation of L-glutamic acid and γ-aminobutyric acid in mice brain. Indian J. Pharm. Sci., 2013, 75 (6), 716–721.
It would be highly desirable to discuss in the text the difference between the authors' analytic methodology and those of these works.

Author Response
Prof. Dr. Markus R. Meyer
Editor in Chief: Metabolites
RE: Manuscript resubmission
Manuscript Number: metabolites-2137299
February 7, 2023
Dear Editor,
Please find enclosed an electronic copy of our manuscript entitled, “Sensitivity Intensified Ninhydrin-based Chromogenic System by Ethanol-Ethyl Acetate: Application to Relative Quantitation of GABA” by Li et al. that we are submitting for consideration of publication in Metabolites. This is a resubmission of an earlier manuscript metabolites-2137299. Thank you very much for your kind letter, along with the constructive comments of the Reviewers. We have thoroughly considered all the comments and revised our manuscript.
We deeply appreciate your consideration of our manuscript. If you have any queries, please don’t hesitate to contact me at the address below.
Yours sincerely,
On behalf of all co-authors,
Jinheng Fu
235# East Nanjing Road, Nanchang 330047, PR China
Phone: +86 79 137 0708 7088
Fax: +86 791 88333708
Email: fujinheng@ncu.edu.cn
Cc. Co-authors
Encl: Manuscript by Li et al.
We sincerely thank you for your constructive comments on our manuscript. A point-by-point response is listed below. We are happy to further edit the manuscript based on your future comments.
Comment 1: As for the readers, the authors’ choice of 40% aqueous ethanol is not obvious or indisputable from the text (see Figure 1), since a 40% ethanol solution diluted with ethyl acetate may be caused a precipitation of some components (or reagents) of the analytical mixture. Moreover, the choice of 60% aqueous ethanol seemed to be more justified, since the absorption value of 60% aqueous ethanol solution was almost three times higher than that of 40% (Figure 1).
Response: Thank you for the comprehensive consideration. We agree that the solvent may cause a precipitation of some component(s). However, the SINICS method is sensitive, a sample generally needs to be diluted prior to assay. For instance, in the validation experiments of this work, we diluted the fermentation samples by 400 times. In addition, the volume ratio of sample to the reacted system is very low (0.1 mL: 3 mL = 3.3%) (Lines 108-109). Therefore, the component(s) is not easy to precipitate.
As you pointed out, 60% aqueous ethanol solution showed a much higher absorption value (A570) than 40% ethanol. However, given that another enhancer, ethyl acetate will also be incorporated into the system, 40% ethanol was thus suggested here (Lines 151-152). Figure 3 indicates that 40:25 of ethanol-ethyl acetate had a higher A570 than 65% ethanol. Moreover, 40:25 of ethanol-ethyl acetate maximized color development of GABA, while did not maximize that of L-glutamic acid. Therefore, we firstly selected 40% ethanol and then supplemented 25% ethanol-ethyl acetate (Lines 189-195).
Comment 2: Although the authors claimed that "All the chromogenic reactions were completed within around 60 min" (part 3.1.1), Figure 1 shows this is not so clear (the absorption value does not yet reach its maximum for this duration). Therefore, calculating the GABA concentration, a certain multiplying factor could be used. The same is true for choosing 25% ethyl acetate instead of 30-35% (see Figure 2).
Response: Thank you for this comment. The reactions stated with "All the chromogenic reactions were completed within around 60 min" were performed under unoptimized conditions. Before we formally began to conduct the optimization experiments, we had done the pre-experiments and known that the optimized SINICS reagent could complete the chromogenic reaction within around 15 min. In fact, although the optimized reaction was indeed completed within 15 min (Figures 6 and 10), we still suggest a reaction time of 30 min (Lines 226-227 and 378).
Comment 3: What is the role of ethyl acetate in the chromogenic reaction? Why does ethyl acetate (according to the authors’ statement) promote "amino acid-ninhydrin reaction"? I can find no explanation in the text. Figure 3 rather refutes the requirement of the ethyl acetate addition, but indicates the need to use a higher concentration of ethanol.
Response: Thank you for this comment. We apologize that the exact role of ethyl acetate is currently unclear. A member of our team is now investigating the role of this reagent.
In Figure 3, the total dosage of ethanol plus ethyl acetate was 65%, but their ratios (35:30, 40:25, 45:20, 50:15, 55:10, 60:5, and 65:0) were various. We expect to enhance A570 of GABA while lower that of L-glutamate as possible. This is helpful to improve the accuracy of GABA analysis, as the higher A570 of GABA is or the lower A570 of L-glutamate is, the less interference of L-glutamate on the assay. The 40:25 of ethanol-ethyl acetate maximized color development of GABA (red curve in Figure 3A), while did not maximize that of L-glutamic acid (red curve in Figure 3B). However, the 65% ethanol maximized A570 values of both GABA and L-glutamate (blue curves in Figures 3A and 3B). Certainly, the maximization of 40:25 ethanol-ethyl acetate on coloring GABA is not so significant. From this aspect, we think that your opinion is very reasonable. We will discuss this to guide our future work.
Comment 4: Regarding to the temperature effect, the authors suggest to use 70°C, although 60°C is obviously better (see Figure 4).
Response: Thank you for this suggestion. At the beginning, it was difficult to decide whether to use 60°C or 70°C. Finally, 70°C was selected because the reaction can be completed quickly at this temperature, and the absorbance value is also satisfactory. Your instructive suggestion provides a good reference for our future work.
Comment 5: Why do the authors believe that pH 5.0 is better than pH 4.5? Over 120 minutes, the absorption values are always higher for pH 4.5 than that for pH 5.0 (see Figure 5).
Response: Thanks very much. At pH 4.5, the stability of chromophore was poorer than that at pH 5.0. In case that many samples are needed to be determined, this instability may result in an inaccurate assay. Based on this consideration, we suggest the pH to be 5.0.
Comment 6: There is a mistype in Abstract: "The kit increases color development sensitivity of L-glutamic acid and GABA, with the reduced detection limit from 20 mM to 5 mM and 20 mM to 5 mM, respectively." According to the information for GABA in part 3.3, the reduced detection limit was from 200.0 mM to 20.0 mM.
Response: Thank you so much for your carefulness. We have revised this accordingly as follows. The kit increased color development sensitivity of L-glutamic acid and GABA, with the detection limits being reduced from 20 mM and 200 mM to 5 mM and 20 mM, respectively (Lines 15-16 and 268-269).
Comment 7: There are two relevant articles for the subject studied:
- a) Y. Zeng, W. Cai, X. Shao, Quantitative analysis of 17 amino acids in tobacco leaves using an amino acid analyzer and chemometric resolution. Separ. Sci., 2015, 38 (12), 2053–2058. DOI: 10.1002/jssc.201500090
- b) J.S. Sancheti, M.F. Shaikh, P.F. Khatwani, S.R, Kulkarni, S. Sathaye, Development and validation of a HPTLC method for simultaneous estimation of L-glutamic acid and γ-aminobutyric acid in mice brain. Indian J. Pharm. Sci., 2013, 75 (6), 716–721.
It would be highly desirable to discuss in the text the difference between the authors' analytic methodology and those of these works.
Response: Thank you for this comment. We have read the two papers and learnt a lot. The difference has been discussed (Lines 278-285 and 341-343).
Reviewer 2 Report
This work is the modification of the ninhydrin classical assay for qualitative determination of GABA in the fermentation process of Levilactobacillus brevis from glutamate. This modification is presented as an easy-handing kit termed SINICS (sensitivity intensified ninhydrin-based chromogenic system). Basically, the improving consisted of the addition of ethanol and ethyl acetate and use of a supposedly optimal pH, incubation temperature and time.
According to paragraph 2.2.1, this modification seems to be an improving of a similar previous kit described at reference [31] using a concentrated acetate buffer without organic solvents.
I think that the optimization of this SINICS in comparison with the previous SINICS described at reference 31 does not is enough to warrant publication in a Journal termed “Metabolites”. This is just a slight modification of a qualitative analytical assay.
Any optimized assay would be compared with other existing methods, but this is not the case. For instance, GABA and any other amino acid can be determined with OPA likely with higher sensitivity. OPA is just mentioned in one paragraph of the manuscript, but no comparison is made. The sensitivity of DFO (1,8-diazafluoren-9-one), a substance known since 1989, is clearly superior to that of ninhydrin for the visualization of latent fingerprints. Others papers for comparison are David L Jones, Andrew G Owen, John F Farrar, Simple method to enable the high resolution determination of total free amino acids in soil solutions and soil extracts, Soil Biology and Biochemistry, 34(12), 2002; or the reference 39, Shih-Wen Sun, Yi-Cheng Lin, Yih-Ming Weng, Min-Jane Chen, Efficiency improvements on ninhydrin method for amino acid quantification, Journal of Food Composition and Analysis, 19, 2006, 112-117.
Interference of other amino acids is not studied, so that the SINICS can be only used for samples with one amino acid. Even in the example, the absorbance at 570 nm is a sum of the contribution of glutamate remaining and GABA fermented. Instead of that, one of the conclusions of the work is that the SINICS method has potential in assaying other amino acids. The use of ninhydrin for amino acid determination does not need to be rediscovered. The versatility is obvious.
Most of the factors introduced for optimization are not surprising. Aside the referred assay described at ref. 31, there are a lot of variations of the original ninhydrin assay using ethanol or acetone for improving the ninhydrin and solubility and the products of the amino acid-ninhydrin reaction, heat and so on. These factors are described in the papers above mentioned.
Minor points
The description of the methods contains numerous ambiguities and trivial points. Some examples:
is seed medium the same one that than fermentation medium?
The concentration of the acetate buffer changes in different paragraphs of the manuscript without justification.
The reason to use ethanol concentration <70% seems to be naïve, as it is stated that pipetting a high concentration (≥ 70%) of ethanol solution is a problem because it will retain on the inner wall of pipet.
The manuscript proposes that methylcellulose solution cannot be used in accelerating amino acid-ninhydrin reactions because of the “toxicity of methylcellulose”. This is totally wrong. Reagents of an assay should not be ingested, but anyway, methylcellulose is allergen-free, non-toxic, colorless, and odorless, making it an excellent gelling agent for various foods. Most methylcellulose will pass through the body undigested.
A solution of 0.2 M monosodium L-glutamate saturated by L-glutamic acid is senseless. The equilibrium between glutamate and glutamic acid depends on the pH.
One of the conclusions of the manuscript is the high sensitivity of the assay, but the proposed limit from 5 to 20 mM is not high sensitivity. Other reagents mentioned above are more sensitive than ninhydrin.
Author Response
Prof. Dr. Markus R. Meyer
Editor in Chief: Metabolites
RE: Manuscript resubmission
Manuscript Number: metabolites-2137299
February 7, 2023
Dear Editor,
Please find enclosed an electronic copy of our manuscript entitled, “Sensitivity Intensified Ninhydrin-based Chromogenic System by Ethanol-Ethyl Acetate: Application to Relative Quantitation of GABA” by Li et al. that we are submitting for consideration of publication in Metabolites. This is a resubmission of an earlier manuscript metabolites-2137299. Thank you very much for your kind letter, along with the constructive comments of the Reviewers. We have thoroughly considered all the comments and revised our manuscript.
We deeply appreciate your consideration of our manuscript. If you have any queries, please don’t hesitate to contact me at the address below.
Yours sincerely,
On behalf of all co-authors,
Jinheng Fu
235# East Nanjing Road, Nanchang 330047, PR China
Phone: +86 79 137 0708 7088
Fax: +86 791 88333708
Email: fujinheng@ncu.edu.cn
Cc. Co-authors
Encl: Manuscript by Li et al.
We sincerely thank you for your constructive comments on our manuscript. A point-by-point response is listed below. We are happy to further edit the manuscript based on your future comments.
Reviewer 2
Major points
Comment 1: According to paragraph 2.2.1, this modification seems to be an improving of a similar previous kit described at reference [31] using a concentrated acetate buffer without organic solvents. I think that the optimization of this SINICS in comparison with the previous SINICS described at reference 31 does not is enough to warrant publication in a Journal termed “Metabolites”. This is just a slight modification of a qualitative analytical assay.
Response: Thank you for this comment. The study principally aims to provide a convenient strategy (SINICS) to relatively quantitate GABA in some biological samples, thus enhancing the work efficiency. The optimization improved the sensitivity of the SINICS.
Comment 2: Any optimized assay would be compared with other existing methods, but this is not the case. For instance, GABA and any other amino acid can be determined with OPA likely with higher sensitivity. OPA is just mentioned in one paragraph of the manuscript, but no comparison is made. The sensitivity of DFO (1,8-diazafluoren-9-one), a substance known since 1989, is clearly superior to that of ninhydrin for the visualization of latent fingerprints. Others papers for comparison are David L Jones, Andrew G Owen, John F Farrar, Simple method to enable the high resolution determination of total free amino acids in soil solutions and soil extracts, Soil Biology and Biochemistry, 34(12), 2002; or the reference 39, Shih-Wen Sun, Yi-Cheng Lin, Yih-Ming Weng, Min-Jane Chen, Efficiency improvements on ninhydrin method for amino acid quantification, Journal of Food Composition and Analysis, 19, 2006, 112-117.
Response: Thank you for providing the valuable information. We have read the relevant references and learnt a lot. The relative comparisons have been supplemented.
Relative to ninhydrin, o-phthalaldehyde and 1,8-diazafluoren-9-one are the two sensitive reagents. Jones et al. has established a spectrofluorometric method based on o-phthalaldehyde, for rapidly and sensitively quantitating total free amino acids in soil. However, o-phthalaldehyde has been more used to derivatize amino acids in the methods with separation step(s), such as HPLC or amino acid analyzer. While 1,8-diazafluoren-9-one is a frequently-used visualizer of latent fingerprints in forensics. When analyzing some biological samples enriched in amino acid(s), such as fermentative ones, ninhydrin-based colorimetry is still rather popular (Lines 278-285).
In this study, we developed a ninhydrin-based colorimetric method (SINICS) for determining GABA of fermentative samples, which was validated by the OPA-based HPLC, suggesting the SINICS could explicate the GABA fermentation process.
Comment 3: Interference of other amino acids is not studied, so that the SINICS can be only used for samples with one amino acid. Even in the example, the absorbance at 570 nm is a sum of the contribution of glutamate remaining and GABA fermented. Instead of that, one of the conclusions of the work is that the SINICS method has potential in assaying other amino acids. The use of ninhydrin for amino acid determination does not need to be rediscovered. The versatility is obvious.
Response: Thanks very much. Ninhydrin can effectively react with all the amino acids. Therefore, the interference of other amino acids is inevitable. We have specified that the SINICS shows potential only in screening amino acids-producing microbes and optimizing/monitoring their fermentation processes (Lines 287-321). These works are frequently confronted with the arduous task of analyzing numerous samples (Lines 355-358). The SINICS has been developed to enhance the efficiency of analysis. After deducting the background control, the net A570 could represent the content of the amino acid. The results gained by the SINICS cannot reflect the real content of an amino acids but can represent its variation trend. We have detailed the application of the SINICS as follows.
Up to date, any ninhydrin-based spectrophotometric method cannot discriminate a specific amino acid from a pool of amino acids, due to the fact that ninhydrin can easily react with any amino acids. In other words, the amino acid amount determined by ninhydrin is total. Herein, we raise a concept that the net value of total A570 minus background is employed as the indication of GABA content of a sample. Obviously, the key to using SINICS is to ensure that all samples have an approximate level of background, so that the net A570 value can correctly reflect the GABA content (Lines 287-293).
With the proceeding of fermentation, dissolved L-glutamic acid was gradually converted into GABA, but was simultaneously compensated by the synchronous dissolution of powder substrate. As a result, the concentration of GABA was gradually increased with the fermentation, while that of dissolved L-glutamic acid was almost invariable before the powder was depleted. At the later stage of fermentation, owing to the depletion of L-glutamic acid powder, the level of dissolved L-glutamic acid may decrease. As stated above, in SINICS analysis, GABA content is directly expressed by subtracting A570 of the background control. It is pivotal to guarantee all samples have a similar background, so as to make the net A570 values comparable. To achieve this, we saturated the samples without visible solid substrate with L-glutamic acid powder prior to SINICS analysis. Another key point is that all samples should be diluted at the same multiple (Lines 300-310).
In this work, all the samples were diluted by 400 folds with ultrapure water before SINICS assay. The fresh fermentation medium (containing 0.2 M monosodium L-glutamate) saturated with L-glutamic acid should be used as the background control. However, the reading of 400-fold dilution of the medium without L-glutamate was negligible (< 0.05). For convenience, the aqueous solution of 0.2 M monosodium L-glutamate saturated by L-glutamic acid powder was selected as the control in this work. All the samples also underwent HPLC analysis, as depicted in the Materials and Methods section. As shown in Figure 9, the plot of GABA production determined by SINICS well followed that determined by HPLC, suggesting SINICS could explicate the GABA fermentation process. In fact, the SINICS kit has been successfully used to optimize the whole-cell catalysis process of L. brevis CD0817 for the production of GABA (Lines 311-321).
Although ninhydrin is universal to all the amino acids, the SINICS is new reagent, and naturally we validated its application to other amino acids.
Comment 4: Most of the factors introduced for optimization are not surprising. Aside the referred assay described at ref. 31, there are a lot of variations of the original ninhydrin assay using ethanol or acetone for improving the ninhydrin and solubility and the products of the amino acid-ninhydrin reaction, heat and so on. These factors are described in the papers above mentioned.
Response: Thank you for this comment. We have cited these references in our manuscript and discussed them (Lines 153-159, 166-168, and 191-195). We are happy to further edit our paper, based on your comments in the future.
Minor points
Comment 1: is seed medium the same one that than fermentation medium?
Response: Thank you for this comment. The two media are different, which has been specified in the text (Lines 111-118). We have checked the whole manuscript to remove the other ambiguities and trivial points.
Comment 2: The concentration of the acetate buffer changes in different paragraphs of the manuscript without justification.
Response: Thanks a lot. Before the section “3.1.6 Effects of sodium acetate concentration”, the concentration of the acetate buffer used in SINICS was 0.8 M (pH 5.8); after that, the 0.2 M (pH 5.0) of acetate buffer was used. The 20.0 mM sodium acetate buffer (pH 7.3) was used to derivatize GABA in HPLC.
Comment 3: The reason to use ethanol concentration <70% seems to be naïve, as it is stated that pipetting a high concentration (≥ 70%) of ethanol solution is a problem because it will retain on the inner wall of pipet.
Response: Thank you for this suggestion. We have deleted this statement.
Comment 4: The manuscript proposes that methylcellulose solution cannot be used in accelerating amino acid-ninhydrin reactions because of the “toxicity of methylcellulose”. This is totally wrong. Reagents of an assay should not be ingested, but anyway, methylcellulose is allergen-free, non-toxic, colorless, and odorless, making it an excellent gelling agent for various foods. Most methylcellulose will pass through the body undigested.
Response: Thank you for this correction. We have deleted this statement.
Comment 5: A solution of 0.2 M monosodium L-glutamate saturated by L-glutamic acid is senseless. The equilibrium between glutamate and glutamic acid depends on the pH.
Response: Thanks very much. A solution of 0.2 M monosodium L-glutamate saturated by L-glutamic acid was used as the background control.
The fermentation medium contained 0.2 M monosodium L-glutamate; and was added solid L-glutamic acid at 650 g/L just before the fermentation. The added substrate mainly existed in the form of powder due to its low solubility (about 10 g/L) (Lines 295-298). With the proceeding of fermentation, dissolved L-glutamic acid was gradually converted into GABA, but was simultaneously compensated by the synchronous dissolution of powder substrate. As a result, the concentration of GABA was gradually increased with the fermentation, while that of dissolved L-glutamic acid was almost invariable before the powder was depleted. At the later stage of fermentation, owing to the depletion of L-glutamic acid powder, the level of dissolved L-glutamic acid may decrease. As stated above, in SINICS analysis, GABA content is directly expressed by subtracting A570 of the background control. It is pivotal to guarantee all samples have a similar background, so as to make the net A570 values comparable. To achieve this, we saturated the samples without visible solid substrate with L-glutamic acid powder prior to SINICS analysis (Lines 300-306).
Comment 6: One of the conclusions of the manuscript is the high sensitivity of the assay, but the proposed limit from 5 to 20 mM is not high sensitivity. Other reagents mentioned above are more sensitive than ninhydrin.
Response: Yes. OPA and DFO are more sensitive than ninhydrin in analyzing amino acids. However, ninhydrin is still an attractive reagent in the analysis of fermentation samples. In our opinion, an appropriate reagent should be selected according to the specific analysis scenario.

Reviewer 3 Report
Li et al. have developed a sensitivity-intensified ninhydrin-based chromogenic system by ethanol-ethyl acetate for its application to relative quantitation of GABA. This study is good attempt toward relative quantitation of GABA and the analysis parameters are methodically optimized. However, there are some concerns as listed below and all which need to be addressed for a possible publication in Metabolites.
1. Abstract – “…with the reduced detection limit from 20 mM to 5 mM and 20 mM to 5 mM, respectively” should be rewritten as “..with the reduced detection limit being 20 mM to 5 mM for both”.
2. Abstract – the sentence “The chromophore maintains … colorimetric analysis” should be rewritten as “The chromophore was stable for at least 2 h at room temperature, which was sufficient for routine colorimetric analysis.”
3. Keywords – “Ninhydrin-based chromogenic reaction” should be “Ninhydrin-based chromogenic system”, “absorbance at 570 nm” should be “colorimetry analysis”.
4. For any method development study, (1) the linear dynamic range of amino acid concentration in which the calibration curve shows linearity, (2) recovery of spiked GABA, (3) reproducibility of method by performing the analysis within a day (intra-day variability) and between days (interday variability), all of these need to be determined. If the authors have done, then include in the manuscript (abstract, methods, results & discussion).
5. Methods – All the purchase details of chemicals/reagents and instruments/equipment/kits should be provided as state, city and country in the case of USA as well as city and country in the case of other countries. Also, for the second instance of same vendor/company’s mention, the authors can simply mention the company name.
6. Section 2.2.4 – what is the mobile phase used? Is the HPLC run by isocratic or gradient mobile phase condition? If isocratic, then provide the initial and final percentage of mobile phase. If gradient, then provide the percentage of mobile phase at different times.
7. Section 2.2.4 – the sentence “In this work … 3 times” should be rewritten as “In this work, all the experiments were repeated at least 3 times”.
8. Section 2 – a statistical analysis sub-section at the end of section 2 should be included to indicate the number of experimental replicates and statistical software, version and purchase details along with the statistical method used for analysis of significance (for example ANOVA or Duncan, student t test etc.)
9. Section 3.5 – “versality” should be corrected as “Versatility”.
10. While the authors claim the versatility of method for analysis of other amino acids as well, how would you justify the developed method to be selective for GABA, as the content determined by this method would be a total amino acid content instead of only GABA. Please clarify and provide suitable explanation/justification.
11. How about the selectivity of this developed method in the presence of other interfering substances in sample matrix?
12. Please mention the limitations of the development method and specify what are the other influencing parameters they/others should evaluate in the future study.
13. It is important to provide the full form of all abbreviations at the first instance only and be abbreviated thereafter.
14. Also, the full form of abbreviations used in the tables and figures should be provided in the respective table footnote and figure caption.
Author Response
Prof. Dr. Markus R. Meyer
Editor in Chief: Metabolites
RE: Manuscript resubmission
Manuscript Number: metabolites-2137299
February 7, 2023
Dear Editor,
Please find enclosed an electronic copy of our manuscript entitled, “Sensitivity Intensified Ninhydrin-based Chromogenic System by Ethanol-Ethyl Acetate: Application to Relative Quantitation of GABA” by Li et al. that we are submitting for consideration of publication in Metabolites. This is a resubmission of an earlier manuscript metabolites-2137299. Thank you very much for your kind letter, along with the constructive comments of the Reviewers. We have thoroughly considered all the comments and revised our manuscript.
We deeply appreciate your consideration of our manuscript. If you have any queries, please don’t hesitate to contact me at the address below.
Yours sincerely,
On behalf of all co-authors,
Jinheng Fu
235# East Nanjing Road, Nanchang 330047, PR China
Phone: +86 79 137 0708 7088
Fax: +86 791 88333708
Email: fujinheng@ncu.edu.cn
Cc. Co-authors
Encl: Manuscript by Li et al.
We sincerely thank you for your constructive comments on our manuscript. A point-by-point response is listed below. We are happy to further edit the manuscript based on your future comments.
Comment 1: Abstract – “…with the reduced detection limit from 20 mM to 5 mM and 20 mM to 5 mM, respectively” should be rewritten as “..with the reduced detection limit being 20 mM to 5 mM for both”.
Response: Thank you for this correction. This has been accordingly revised. The sentence has been revised as: The kit increased color development sensitivity of L-glutamic acid and GABA, with the detection limits being reduced from 20 mM and 200 mM to 5 mM and 20 mM, respectively. We are happy to further edit our paper based on your comments in the future.
Comment 2: Abstract – the sentence “The chromophore maintains … colorimetric analysis” should be rewritten as “The chromophore was stable for at least 2 h at room temperature, which was sufficient for routine colorimetric analysis.”
Response: Yes. This has been accordingly revised.
Comment 3: Keywords – “Ninhydrin-based chromogenic reaction” should be “Ninhydrin-based chromogenic system”, “absorbance at 570 nm” should be “colorimetry analysis”.
Response: Thank you for this revision. These key words have been accordingly revised.
Comment 4: For any method development study, (1) the linear dynamic range of amino acid concentration in which the calibration curve shows linearity, (2) recovery of spiked GABA, (3) reproducibility of method by performing the analysis within a day (intra-day variability) and between days (interday variability), all of these need to be determined. If the authors have done, then include in the manuscript (abstract, methods, results & discussion).
Response: Thank you for this comment. Before starting this work, we considered these items you mentioned. These items are really needed in the absolute quantitation techniques with separation step(s) like HPLC. However, the SINICS method directly uses A570 to express the concentration of amino acid. And all SINICS assays are performed in parallel, that is, 2.9 mL aliquots of the SINICS master mix are added into 0.1 mL samples, then incubated in a water bath, subsequently the A570 values are read and compared. Therefore, in our view, they may not be mandatory in this study. Of course, it will be better if these items are determined. Your constructive comment will improve our next works.
Comment 5: Methods – All the purchase details of chemicals/reagents and instruments/equipment/kits should be provided as state, city and country in the case of USA as well as city and country in the case of other countries. Also, for the second instance of same vendor/company’s mention, the authors can simply mention the company name.
Response: Thank you for this comment. We have supplemented these items accordingly.
Comment 6: Section 2.2.4 – what is the mobile phase used? Is the HPLC run by isocratic or gradient mobile phase condition? If isocratic, then provide the initial and final percentage of mobile phase. If gradient, then provide the percentage of mobile phase at different times.
Response: Thank you for this comment. The mobile phase was consisted of solvent A and solvent B at a volume ratio of 4:1. Solvent A was prepared by mixing 9.8 mL of 20.0 mM sodium acetate buffer (pH 7.3) and 0.2 mL triethylamine. Solvent B was acetonitrile (Lines 91-94). The column was eluted by an isocratic mobile phase at a flow rate of 0.8 mL/min, an oven temperature of 30 °C, and a detection wavelength of 338 nm (Lines 133-135).
Comment 7: Section 2.2.4 – the sentence “In this work … 3 times” should be rewritten as “In this work, all the experiments were repeated at least 3 times”.
Response: Thank you for this correction. The sentence has been accordingly revised.
Comment 8: Section 2 – a statistical analysis sub-section at the end of section 2 should be included to indicate the number of experimental replicates and statistical software, version and purchase details along with the statistical method used for analysis of significance (for example ANOVA or Duncan, student t test etc.)
Response: Thank you for this comment. This has been accordingly provided (Lines137-140).
Comment 9: Section 3.5 – “versality” should be corrected as “Versatility”.
Response: Yes. This word has been corrected.
Comment 10: While the authors claim the versatility of method for analysis of other amino acids as well, how would you justify the developed method to be selective for GABA, as the content determined by this method would be a total amino acid content instead of only GABA. Please clarify and provide suitable explanation/justification.
Response: Thank you for this comment. This has been clarified as follows.
Up to date, any ninhydrin-based spectrophotometric method cannot discriminate a specific amino acid from a pool of amino acids, due to the fact that ninhydrin can easily react with any amino acids. In other words, the amino acid amount determined by ninhydrin is total. Herein, we raise a concept that the net value of total A570 minus background is employed as the indication of GABA content of a sample. Obviously, the key to using SINICS is to ensure that all samples have an approximate level of background, so that the net A570 value can correctly reflect the GABA content (Lines 287-293).
With the proceeding of fermentation, dissolved L-glutamic acid was gradually converted into GABA, but was simultaneously compensated by the synchronous dissolution of powder substrate. As a result, the concentration of GABA was gradually increased with the fermentation, while that of dissolved L-glutamic acid was almost invariable before the powder was depleted. At the later stage of fermentation, owing to the depletion of L-glutamic acid powder, the level of dissolved L-glutamic acid may decrease. As stated above, in SINICS analysis, GABA content is directly expressed by subtracting A570 of the background control. It is pivotal to guarantee all samples have a similar background, so as to make the net A570 values comparable. To achieve this, we saturated the samples without visible solid substrate with L-glutamic acid powder prior to SINICS analysis. Another key point is that all samples should be diluted at the same multiple (Lines 300-310).
In this work, all the samples were diluted by 400 folds with ultrapure water before SINICS assay. The fresh fermentation medium (containing 0.2 M monosodium L-glutamate) saturated with L-glutamic acid should be used as the background control. However, the reading of 400-fold dilution of the medium without L-glutamate was negligible (< 0.05). For convenience, the aqueous solution of 0.2 M monosodium L-glutamate saturated by L-glutamic acid powder was selected as the control in this work. All the samples also underwent HPLC analysis, as depicted in the Materials and Methods section. As shown in Figure 9, the plot of GABA production determined by SINICS well followed that determined by HPLC, suggesting SINICS could explicate the GABA fermentation process. In fact, the SINICS kit has been successfully used to optimize the whole-cell catalysis process of L. brevis CD0817 for the production of GABA (Lines 311-321)
Comment 11: How about the selectivity of this developed method in the presence of other interfering substances in sample matrix?
Response: Thank you for this comment. The GABA content is directly expressed by subtracting A570 of the background control. The samples from different fermentation stages had similar backgrounds. The dilution of the samples further overcame the background (difference). In addition, the volume ratio of sample to the reacted system is very low (0.1 mL: 3 mL = 3.3%) (Lines 109-110), which also minimized the background (difference). As a result, the backgrounds of the samples were negligible or almost the same.
In this work, all the samples were diluted by 400 folds with ultrapure water before SINICS assay. The fresh fermentation medium (containing 0.2 M monosodium L-glutamate) saturated with L-glutamic acid should be used as the background control. However, the reading of 400-fold dilution of the medium without L-glutamate was negligible (< 0.05). For convenience, the aqueous solution of 0.2 M monosodium L-glutamate saturated by L-glutamic acid powder was selected as the control in this work (Lines 311-316).
Comment 12: Please mention the limitations of the development method and specify what are the other influencing parameters they/others should evaluate in the future study.
Response: Thank you for this comment. The limitations have been added in the manuscript.
Like the other spectrophotometric methods, the limitation of SINICS is that this procedure cannot absolutely quantitate target amino acid, because it lacks the ability to distinguish amino acids. We recommend the combination of the SINICS and an absolute quantitative method, that is, the former is exploited to complete optimization/monitoring, and the latter to measure the optimized content of amino acid. Zeng et al. has developed an amino acid analyzer protocol for simultaneous quantifying 17 amino acids in tobacco leave, in which amino acids were derivatized with ninhydrin. Sancheti et al. has proposed a ninhydrin-based high performance thin layer chromatographic method to simultaneously assay L-glutamic acid and GABA in mice brain. These accurate methods are suitable for determining the final content of amino acid after SINICS optimization (Lines335-345).
And the other influencing parameters have been specified.
It would be interesting to identify a reagent that can fortify the color development of target amino acid, because this reagent may improve the reliability of analysis by reducing the interference of the other amino acids and sample matrix. Figure 3 shows that 40:25 of ethanol-ethyl acetate maximized the color development of GABA, but did not maximize that of L-glutamic acid, suggesting the possibility of obtaining such a reagent (Lines 346-350).
Comment 13: It is important to provide the full form of all abbreviations at the first instance only and be abbreviated thereafter.
Response: Yes, we have thoroughly checked the manuscript for this aim.
Comment 14: Also, the full form of abbreviations used in the tables and figures should be provided in the respective table footnote and figure caption.
Response: Thank you for this comment. These have been accordingly revised.
Reviewer 4 Report
The scope of this manuscript is interesting. The paper presents detailed data on the development of a new method for the quantitative determination of gamma-aminobutyric acid. The work is technically well structured – everything is well organized and clearly described. However, one important aspect has been overlooked.
The authors report on the quantitative analysis of GABA and L-glutamic acid, but there is no detailed information on the validation of the developed method. Apart from the detection limit, other validation parameters are missing. Therefore, more details on the quantification of analytes should be provided. What about concentration range (calibration curve should be shown), determination precision (SD and RSD) and recovery? And what about comparing the results obtained with the HPLC method? For this reason, a new table should be included in the manuscript to show the values for the missing validation parameters.
Abstract: Please confirm that sentence:
“The kit increases color development sensitivity of L-glutamic acid and GABA, with the reduced detection limit from 20 mM to 5 mM and 20 mM to 5 mM, respectively.”
is spelled correctly.
2.1. Materials: Please explain what "ddH2O" means. How is it prepared?
Author Response
Prof. Dr. Markus R. Meyer
Editor in Chief: Metabolites
RE: Manuscript resubmission
Manuscript Number: metabolites-2137299
February 7, 2023
Dear Editor,
Please find enclosed an electronic copy of our manuscript entitled, “Sensitivity Intensified Ninhydrin-based Chromogenic System by Ethanol-Ethyl Acetate: Application to Relative Quantitation of GABA” by Li et al. that we are submitting for consideration of publication in Metabolites. This is a resubmission of an earlier manuscript metabolites-2137299. Thank you very much for your kind letter, along with the constructive comments of the Reviewers. We have thoroughly considered all the comments and revised our manuscript.
We deeply appreciate your consideration of our manuscript. If you have any queries, please don’t hesitate to contact me at the address below.
Yours sincerely,
On behalf of all co-authors,
Jinheng Fu
235# East Nanjing Road, Nanchang 330047, PR China
Phone: +86 79 137 0708 7088
Fax: +86 791 88333708
Email: fujinheng@ncu.edu.cn
Cc. Co-authors
Encl: Manuscript by Li et al.
We sincerely thank you for your constructive comments on our manuscript. A point-by-point response is listed below. We are happy to further edit the manuscript based on your future comments.
Comment 1: The authors report on the quantitative analysis of GABA and L-glutamic acid, but there is no detailed information on the validation of the developed method. Apart from the detection limit, other validation parameters are missing. Therefore, more details on the quantification of analytes should be provided. What about concentration range (calibration curve should be shown), determination precision (SD and RSD) and recovery? And what about comparing the results obtained with the HPLC method? For this reason, a new table should be included in the manuscript to show the values for the missing validation parameters.
Response: Thank you for this comment. Before starting this work, we considered these items you mentioned. These items are really needed in the absolute quantitation techniques with separation step(s) like HPLC. However, the SINICS method directly uses A570 to express the concentration of amino acid. And all SINICS assays are performed in parallel, that is, 2.9 mL aliquots of the SINICS master mix are added into 0.1 mL samples, then are simultaneously incubated in a water bath, subsequently the A570 values are read and compared. Therefore, in our view, they may not be mandatory in this study. Of course, it will be better if these items are determined in this work. Your constructive comment will improve our next works.
The SINICS method has been validated by the HPLC. As shown in Figure 9, the results obtained with SINICS followed a similar trend with those obtained with HPLC.
Comment 2: Abstract: Please confirm that sentence:
“The kit increases color development sensitivity of L-glutamic acid and GABA, with the reduced detection limit from 20 mM to 5 mM and 20 mM to 5 mM, respectively.”
is spelled correctly.
Response: Thank you for this correction. This has been revised accordingly. The kit increased color development sensitivity of L-glutamic acid and GABA, with the detection limits being reduced from 20 mM and 200 mM to 5 mM and 20 mM, respectively (Lines 15-16 and 268-269).
Comment 3: 2.1. Materials: Please explain what "ddH2O" means. How is it prepared?
Response: Thank you for this comment. ddH2O means double distilled water. We prepared ddH2O by Milli-Q Ultrapure Water System (Millipore Corporation, Billerica, MA, USA), thus it is more appropriate to call it ultrapure water. We have replaced ddH2O with ultrapure water.

Round 2
Reviewer 2 Report
I recognize the good job made by the authors during the modification of the manuscript. I am grateful for the clarity in the reply letter and I feel that the modifications introduced in the manuscript improved this possible artcle. However, the core and main objetive of the manuscript is just improving the conditions for reaction of ninhidrin with GABA in a special conditions where glutamate is transfomed in GABA. The use of ninhidrin for amino acid detection is well known decades ago. In sum. I leave the final decision to the editor, but I cannot totally change my initial opinion about novelty, significance of content and scientific soudness of the work.
Reviewer 3 Report
The authors have satisfactorily addressed all the comments raised by reviewers and therefore I recommend acceptance of this article for publication in Metabolites.